# Parents’ Experiences of Weighted Blankets’ Impact on Children with Attention-Deficit/Hyperactivity Disorder (ADHD) and Sleep Problems—A Qualitative Study

**DOI:** 10.3390/ijerph182412959

**Published:** 2021-12-08

**Authors:** Ingrid Larsson, Katarina Aili, Jens M. Nygren, Håkan Jarbin, Petra Svedberg

**Affiliations:** 1Department of Health and Care, School of Health and Welfare, Halmstad University, SE-30118 Halmstad, Sweden; jens.nygren@hh.se (J.M.N.); petra.svedberg@hh.se (P.S.); 2Department of Health and Sport, School of Health and Welfare, Halmstad University, SE-30118 Halmstad, Sweden; Katarina.aili@hh.se; 3Child and Adolescent Psychiatry, Department of Clinical Sciences Lund, Lund University, SE-22184 Lund, Sweden; hakan.jarbin@regionhalland.se; 4Child and Adolescent Psychiatry, Region Halland, SE-30185 Halmstad, Sweden

**Keywords:** attention-deficit/hyperactivity disorder (ADHD), experiences, children, intervention, parents, qualitative content analysis, sleep problems, weighted blankets

## Abstract

Sleep disturbances are common among children with attention-deficit/hyperactivity disorder (ADHD). While pharmacological treatment has increased dramatically, parents often prefer non-pharmacological interventions. Research on experiences of weighted blankets and their effect in sleep improvement is scarce. The aim of this study was to explore parents’ experiences of weighted blankets for children with ADHD and sleep problems, and the impact on their children’s sleep. The explorative design was based on qualitative content analysis. Interviews were conducted with a purposeful sample of 24 parents of children with ADHD and sleep problems, after completing a sleep intervention with weighted blankets for 16 weeks. Parents reported that children sleeping with weighted blankets: (1) achieved satisfactory sleep, including improved sleep onset latency, sleep continuity, and sleep routines; (2) achieved overall well-being, including improved relaxation and reduced anxiety; and (3) mastered everyday life, including improved balance in life, family function, and participation in school and leisure activities. This study brings forward novel aspects of the effects of improved sleep among children with ADHD. The findings contribute to the understanding of potential positive effects of an intervention with weighted blankets critical for clinical practice to improve sleep, well-being, and everyday life of children with ADHD and their families.

## 1. Introduction

Attention-deficit/hyperactivity disorder (ADHD) affects approximately 5% of school-age children worldwide [1] and is characterized by impulsivity, inattention, and/or hyperactivity [2]. Children with ADHD have an increased risk of poor health outcomes, compared to healthy peers. The negative behaviors associated with ADHD cause difficulties with interpersonal interactions and educational outcomes, and are likely to be exacerbated by sleep disturbances [3]. The prevalence of sleep disturbances among children with ADHD is high (up to 70%) [4,5], compared to healthy peers (20–30%) [6]. Children with ADHD have been found to have between 30 to 60 min shorter sleep duration and significantly more night awakenings, compared to healthy peers [7,8,9]. Sleep disturbance includes both initiation and maintenance of sleep. The most common sleep disturbances among children with ADHD are: bedtime refusal (e.g., resisting going to bed), difficulties in initiating sleep (e.g., insomnia), difficulties staying asleep (e.g., nighttime awakening), disproportionate daytime sleepiness, and fragmented sleep [3,5].

Sleep is essential for health and well-being and is important for everyday activities [10,11,12]. Among children, poor sleep negatively affects physical and mental health [13,14], quality of life, [15] along with performance [14] and relationships at school [16]. Sufficient duration and sleep quality are associated with better attention, behavior, and cognitive functions, and better physical and mental health among children [17,18]. Satisfactory sleep is thus important for school achievement and transition into adulthood and working life, especially for children with ADHD [19,20]. Further, children’s sleep disturbances have a significant impact on parents’ everyday lives and are associated with higher parental stress, poorer parental mental health, and increased parenting stress [21]. This emphasizes the importance of taking a whole-family perspective when managing sleep problems among children with ADHD [21,22].

The use of pharmacological treatment (e.g., melatonin) for sleep problems among children with ADHD is common and has increased dramatically in the past 10 years, although often with unfavorable side effects. Melatonin is the most common medication for sleep in various neurological conditions and is regarded as safe and effective [23] but might cause morning drowsiness, increased enuresis, headache, dizziness, diarrhea, rash, and hypothermia [24]. However, if there are alternatives, parents prefer non-pharmacological interventions to pharmacological treatment [25]. Interventions with weighted blankets are used as a replacement or supplement to pharmacological treatment for sleep problems [26,27,28]. The use of weighted blankets is based on the theory of sensory integration with assumptions of how the body reacts to different stimuli depending on the sensitivity to stimuli [29,30]. A weighted blanket is considered a sensory-stimulating, cognitive aid that provides a deep pressure on the body (deep pressure touch stimulation). The pressure created by the weight from the weighted blanket leads to a feeling of being enveloped and increasing body awareness [31]. The effectiveness of weighted blankets among children with ADHD is essentially unexplored [26,27,28]. Only two studies (one case-control study, and one intervention study without controls) investigating the effect of weighted blankets among children with ADHD have been found. The studies found positive effects of weighted blankets on sleep, ADHD symptoms, daily level of functioning, and quality of life [32,33]. Only one randomized controlled trial (RCT) investigating the effect of weighted blankets on sleep in children has been found; however, that study only included children with autism spectrum disorders. No positive effects on sleep were seen in the study, although both parents and children favored weighted blankets over their usual blanket [34]. In summary, there is no convincing evidence of the effect [28] or the economic impact [27,35], nor any qualitative studies from children’s and parents’ perspectives on using weighted blankets to improve sleep in children with ADHD [3]. In filling this gap [35], an RCT by our research group is underway to explore the effects of an intervention with weighted blankets on both sleep and everyday life for children with ADHD and sleep problems [36]. By way of this RCT, we will provide new evidence of the efficacy, cost-effectiveness, and experiences of the intervention from children’s and parents’ perspectives.

This qualitative study contributes to increased knowledge on the impact of an intervention with weighted blankets on sleep among children with ADHD from their parents’ perspectives. This type of knowledge is important, both to provide understanding of the intervention as such, and in the development of sustainable interventions using weighted blankets for sleep improvement in clinical practice. The aim of this study was to explore parents’ experiences of a non-pharmacological intervention with weighted blankets for children with ADHD and sleep problems and the impact on their children’s sleep.

## 2. Materials and Methods

### 2.1. Design

The study had an explorative design based on inductive qualitative content analysis, which aims to explore variations by identifying differences and similarities in a text, formulated in categories and themes. Qualitative content analysis is useful for analyzing an individual’s experiences, reflections, or attitudes [37,38]. To ensure trustworthiness, the study is reported following the Consolidated Criteria for Reporting Qualitative Research 32-item checklist [39].

### 2.2. Intervention

This study focused on the parents’ experiences of a sleep intervention with weighted blankets that included children aged 6–15 years old with newly diagnosed ADHD and sleep problems [36]. Diagnoses were based on a detailed written report from school including a SNAP-IV rating scale [40] and an interview with parents and the child performed by a registrar or locum junior doctor utilizing a structured protocol. The decision was always discussed and confirmed by a consultant child psychiatrist during the diagnostic session. The consultant could choose to interview the child and parents as needed but an oral report was mandatory and mostly sufficient. The sleep intervention was based on a comparison of fiber-weighted blankets and control blankets–i.e., fiber blankets without additional weight. The weighted blanket and the control blanket had the same design, so that the weight was the only aspect that distinguished them. The fibers used in the weighted blanket consisted of polyester filaments. The weight of each blanket was individually tailored (weight between 6 and 10 kg) for the children based on age, sex, height, weight, degree of sleep problems, and subtypes of ADHD, in accordance with the use in the clinical practice of two, independent, experienced occupational therapists. The children were randomized to start either with a weighted blanket or with a control blanket and used each blanket for four weeks and then changed to the other blanket (a crossover design). After eight weeks, the child was to choose one of the blankets and use this blanket for an additional eight weeks, thus the duration of the intervention was 16 weeks.

### 2.3. Participants

A purposeful sample was carried out among the parents in the sleep intervention. The inclusion criteria were: a parent of a child with ADHD and sleep problems, the child had completed the RCT, the child chose to continue with the weighted blanket for the 8-week follow-up. The participants were strategically selected to achieve variation in sex, age, civil status, educational level, employment, place of residence. Twenty-six parents of the children in the sleep intervention [36] were approached at the end of the intervention and asked to participate in an interview about their experiences of using weighted blankets. A total of 24 parents accepted the invitation and participated (Table 1).

### 2.4. Data Collection

Data collection took place between March and September 2020. The interviews were performed by the first author (IL) in an undisturbed room at Halmstad University. Individual interviews were performed with 18 parents, and three parent couples (six parents) chose to be interviewed in pairs. The interviews were conducted in Swedish. The interviews started with questions about the parents’ experiences of the two different blankets and then in-depth questions about their experiences of the weighted blankets. The following questions were asked: “What are your experiences of the two blankets your child has tested?”, “What are your experiences from your child’s use of a weighted blanket?”, “Has your child’s sleep changed with the use of a weighted blanket? If yes, in which way?”, “Has your child’s everyday life changed with the use of a weighted blanket? If yes, can you describe how?”, and “Has your family life changed with the use of a weighted blanket, and, if so, can you describe how?” To obtain comprehensive descriptions, follow-up questions were posed, such as: “What expectations did you have of each of the two blankets?”, and “How have your expectations been met?” Probing questions were used, such as: “Please, tell me more about…”, “What do you mean?” or “What do you have in mind, when you say…?” Two pilot interviews were conducted to test the questions and, because adjustments were not required, these interviews were included in the study. The interviews lasted between 35 and 64 min, with a median of 53 min and a total interview time of 17 h and 12 min. All interviews were digitally recorded and transcribed verbatim.

### 2.5. Data Analysis

Data were analyzed in a latent qualitative content analysis [37,41]. The interviews were listened to and transcribed. The transcriptions were read through repeatedly, to become familiar with the content and obtain a sense of the whole. Phrases containing content relevant to the aim were identified and extracted in meaning units. The meaning units were abstracted and coded. The codes were compared on the basis of similarities and differences and grouped into nine sub-categories and three categories that reflected the central and manifest content in the interviews. The analysis was conducted by the first (IL) and last authors (PS). There were continuous discussions and reflections between all the authors to achieve a consensus of the analysis.

### 2.6. Ethical Considerations

The study was approved by the Swedish Ethical Review Authority (no. 2019-02158) and conforms to the principles outlined in the Declaration of Helsinki [42]. The study fulfilled requirements on research: information, consent, confidentiality, and safety of the participants and was guided by the ethical principles of autonomy, beneficence, non-maleficence, and justice [43]. The parents received written and oral information from the first author (IL) and provided informed consent in writing. The information contained voluntary participation and the possibility of withdrawing at any time without having to give a reason.

## 3. Results

Parents experienced that, during the intervention, their children achieved satisfactory sleep, overall well-being, and mastered everyday life when they used the weighted blankets. Sleeping with weighted blankets resulted in satisfactory sleep, including improved sleep onset latency, sleep continuity, and sleep routines. The overall well-being of the children was improved, including improved relaxation, and reduced anxiety. Better sleep with weighted blankets contributed to an improved mastering of everyday life, including balance in life, family function, and participation in school and leisure activities for children with ADHD and sleep problems (Table 2).

### 3.1. Achieving Satisfactory Sleep

Satisfactory sleep was achieved through improved sleep onset latency, sleep continuity, and sleep routines when children used the weighted blanket.

#### 3.1.1. Improved Sleep Onset Latency

The parents described that the children had improved sleep onset latency with the weighted blanket. Before using weighted blankets, children had difficulties calming down, resting, and falling asleep when they lay in bed before nightfall. It was common that children felt worried or restless with difficulties lying still and coming to rest. The weighted blanket gave them a feeling of being embraced and a sense of security, making it easier for the child to fall asleep.


*With the weighted blanket, he does not move as much when he goes to sleep. He becomes calmer and falls asleep faster. He said the weighted blanket is good to hold on to as it hugged him and that turning around and moving became harder and heavier. We think he sleeps better with the weighted blanket. He falls asleep faster and is less unsettled and also sleeps a little longer in the morning.*
(Parent no. 21)

Many of the parents described that the children previously had wanted the parents to be there throughout the whole process of falling asleep, providing comfort by lying next to them, hugging them, and being very close, for example by placing a leg or arm over them, to make them feel safe. When the children instead used the weighted blanket, they acquired a similar embracing effect which allowed them to fall asleep by themselves. The parents further described that some children no longer showed resistance towards going to bed when using the weighted blanket.


*She has just embedded herself (with the weighted blanket) and then it’s just been goodnight. It’s been unbelievable. Before, she could lie in bed for several hours until 11 p.m.... So, at 11 p.m. you could hear, “Mom, Dad.” But since she got the weighted blanket, it was just, “Go to bed”. Before, I had to sit there and have the light turned on. Now she goes upstairs by herself and crawls under and turns off the light. We used to do that every night and we’d have to sit there. We could sometimes sit there for an hour, but it’s completely over and also, it was quite important that both parents would come up and put her to bed. She wanted somehow to know that we were both at home. Now we don’t have to be home. She goes to bed anyway. Yes, even if we’re not at home.*
(Parent no. 19)

Even when children still needed a parent by their side to fall asleep, they fell asleep faster with the weighted blanket. Their children’s ability to stay asleep and not wake up when the parent left the room was further appreciated by parents.


*He always has to sleep with an adult to relax and go to sleep, but he sleeps so lightly that you can’t move and then it’s often during the night that he sleeps so lightly. He has used me as a pacifier blanket. He fidgets and moves and stuff like that. With the weighted blanket, we’ve noticed that sleep initiation has shortened... Then you give him that pressure and put a heavy arm and sometimes I almost lean over him and then, when he has fallen asleep, I put on the weighted blanket and then we can walk out of his room almost without even sneaking. Before, I almost couldn’t even sit up without him jerking on and off and saying, “Mom, mom” because he slept so super lightly and he has always been so easily awakened.*
(Parent no. 20)

There were parents who described that the child did not get the effect from the weighted blanket on the sleep onset latency as they expected, but that the child slept better during the night.


*She might not fall asleep any faster, but she sleeps really, really well.*
(Parent no. 12)

#### 3.1.2. Improved Sleep Continuity

The parents described that the children had improved sleep continuity. They slept more calmly, deeply, and continuously, and thus received more hours of night sleep than before using the weighted blanket. The weighted blanket contributed to a calmer and more restful sleep, in that the children were lying more still and not moving around in the bed as before. This, in turn, led to the children being more relaxed during the night. Previously, the children moved around a lot during sleep; for example, making rapid movements with the legs brushing the wall or moving so much that they could wake up with their head at the foot of the bed and without a blanket. This contributed to the children waking up more during the night, meaning that the parents and/or siblings also woke up. However, the parents described that the children’s sleep was more persistent and continuous with the weighted blanket, resulting in increased sleep duration.


*She seems to be calmer when she sleeps. You’ll pretty much find her in the same position in the morning as when she fell asleep last night. It’s not like this electric whisk. We’ve always called her the electric whisk, if she fell asleep with her head at one end, she was at the other end when she woke up. She has always been like that, but I feel like she is more at peace. But, as she says, she can’t be bothered to move and it makes her lie still. She doesn’t wake up needing to move... She said that with this (weighted blanket) she slept much, much better so she noticed a huge difference with it... She likes it very much and sleeps very well with the heavy blanket.*
(Parent no. 8)

The parents also expressed that the weighted blanket contributed to a more continuous and deeper sleep, without several awakenings during the night. Before using the weighted blanket, some children were easily disturbed during sleep and often went to the bathroom during the night or to the kitchen to drink water or woke up easily from any noise. Some parents described that their child seemed to get a deeper sleep than they had ever had before. Many children slept all night in their own bed with the weighted blanket, without going into their parents’ bedroom.


*He doesn’t wake up. I think it happened once that he got up and then I reacted, but then he needed to wee or go to the bathroom. But it was completely different to the other weeks, so it was normal to get up if you needed to wee or something. But he sleeps all night now... he didn’t before, absolutely not. His sleep was fragmented all the time. He woke up and fell asleep and he called for us many times.*
(Parent no. 5)

However, some parents described that the improved sleep disappeared or diminished during the intervention, when the child had to change to a fiber blanket without weight and that the effect was not fully restored when the child returned to a weighted blanket.


*When she had the weighted blanket in the first period (of the intervention), she slept fantastically well. It was wonderful because she slept in her bed all night. I don’t think she came in to us one single night during that period. The first day we switched to the lighter blanket, she came in to us right away and did so all the time with the lighter blanket. When we switched back to the heavy blanket, I thought, “Oh, good, let’s see if she’s asleep again.” But now she’s slept worse with that one. At some point she’s probably slept in her bed, but she often comes in to us anyway now... Maybe she’s slept a little better with the heavier than with the lighter blanket. Because then she came in as soon as we went to bed. Now she sleeps a bit longer in her own bed... For the first four weeks (with the weighted blanket), she did not once come in to us but slept all the time in her own bed. It has never happened before.*
(Parent no. 14)

Some parents described that the children liked the weighted blanket in the beginning, especially the first few weeks, but then did not think that it had any special effect. Some children also experienced that the blanket was too hot or that it was too lumpy and uncomfortable (not the weight as such) and that, therefore, they did not like the weighted blanket. To prevent the weighted blanket from feeling lumpy and inconvenient, some children preferred to have a soft, thin blanket close to the body and the weighted blanket on top.


*He liked it from the start, but then more and more he didn’t want it. He thought it was lumpy, “rough” he called it. He didn’t find it very comfortable. In the beginning, he liked it. I don’t know what he said about it at the time, but then he still thought it was okay. Now he thinks it’s rough and not nice. I don’t think it’s the weight itself, but more that it’s uncomfortable and lumpy.*
(Parent no. 9)

#### 3.1.3. Improved Sleep Routines

The parents described that the weighted blankets contributed to better sleep routines for their children. When the child felt the weight of the blanket, it was a sign that it was bedtime. The parents described that the routines around betimes were important for the children and the weighted blanket contributed to this process, by giving both more procedure to the routines and a feedback sensation that enforced the routine. Improved sleep routines for bedtime led to the children achieving a normal sleep–wake cycle.


*It is becoming clearer that now we go to bed, to lie down under the blankets. Perhaps it is the clarity that is also beneficial. Now it’s tucked in and we’ve gone to bed. We always pull on the weighted blanket, it’s a natural part of it. He has his routine around this. He has to be on my arm, with his stuffed animal, and on with the blanket, then it’s kind of done. Yes, it’s dark and it’s quiet, and that’s it. Then it’ll take five minutes. He falls asleep in his own bed every night. It’s become more often that it’s in his bed, his weighted blanket. I don’t know, but I feel like the routine has got a little more settled. He likes it. He likes the bed and the blanket and how we arrange it, same procedure every evening.*
(Parent no. 4)

The parents described that the improved sleep routines resulted in the children preferring to stay in bed at bedtime. For some children, the weighted blanket contributed to a feeling of comfort and that the weight from the blanket was appreciated and made them want to stay in bed. For other children, the reason to stay in bed was more the result of convenience as it became more cumbersome to get out of bed due to the weight of the blanket. Before using weighted blankets, some children had a habit of getting out of bed after bedtime, repeatedly coming up with different reasons for not staying in bed and often wandering around the house for no particular reason. This behavior changed for the better with the weighted blankets.


*NN likes this (weighted) blanket very much that she’s had now because we don’t hear her running around at all. She says herself that it just feels hard to get up and move. I don’t even think she wakes up feeling like she needs to get up and go.*
(Parent no. 8)

Some children preferred to use the weighted blankets as an ordinary blanket, whereas others preferred to only use it to apply pressure to some parts of the body, for example, the back, or not to cover some part of the body, for example, the legs. The children who preferred to have their parents close to them experienced that when the blanket embraced them, it functioned as a substitute for the parent.


*I put it behind his back like I was still there. So it’s like he shouldn’t feel much difference. There’s still a pressure there, so I think it’s a good thing it’s going to be a little substitute… that I could replace myself a little bit with this (weighted blanket) so he could still feel a little bit embraced and thus he is lying still and he likes it before.*
(Parent no. 4)

The change of routines supported by the weighted blankets required some patience on the part of the children before they got used to it and figured out individual adaptations. At first, some children found it uncomfortable and expressed a strange feeling. Some initially experienced side effects and that a period of habituation was needed. For example, some children experienced a feeling of confinement, resulting in nightmares about being trapped, but the children did not want to give up and when they got used to the weighted blankets after a few days, they felt secure and calm. They found a new routine and they did not want to change back to a regular blanket.


*Initially, she had nightmares with the weighted blanket, she thought she was trapped... So it probably took three or four nights, I believe, before she felt she could settle down. *
(Parent no. 1)

### 3.2. Experiencing Overall Well-Being

Nights of good sleep with a weighted blanket had a positive effect on the child’s well-being, through improved relaxation and reduced anxiety.

#### 3.2.1. Improved Relaxation

The parents described that the weighted blanket provided a sensation of being touched and embraced and that this provided calm, comfort, and relaxation for the children. The feeling of pressure on the body provided comfort and calm. The calmness influenced the children, both mentally and physically.


*He always wants the heavy blanket on top of him. He feels a bit calmer and somewhat more secure with the (weighted) blanket. He himself says that he feels more relaxed when he lies under it. He likes to be close and that’s probably the feeling he gets... I think he feels closeness and security with the weighted blanket.*
(Parent no. 24)

The children expressed that the blanket hugged them and that it gave them a feeling of comfort. Some children described that they felt a big difference between the weighted and the ordinary blanket and that they loved the weighted one.


*She says herself that it feels like, she calls it a cuddle blanket. It feels like the weighted blanket is hugging me.*
(Parent no. 10)

Some children even wanted to use the weighted blanket when they rested for a while after school. The parents described that the child wanted a relaxing and cozy moment when, for example, they would check their phone.

*When she had that fiber blanket (the light blanket), she went in during the day on top of my bed to rest. Because she wanted to lie under my weighted blanket, because she thought it was nice to relax. Preferably after school and lie there and look into the phone... Just rest and feel a bit relaxed, I think. Yes, she did not sleep nearly as well (when the weighted blanket was removed and the child got the light blanket); then she was more unsettled. She needed to get some rest after school, as I said, then she wanted to lie in my bed and rest under my weighted blanket. It was the weight she wanted*.(Parent no. 12)

#### 3.2.2. Reduced Anxiety

The parents described an overall improved well-being in their children, with reduced anxiety due to the use of weighted blankets and the resulting improved sleep. The previous fear and anxiety of not being able to fall asleep or sleep through the night and the fatigue that they had experienced before had led to anxiety in some children. Reduced anxiety also allowed children to focus, reduced their hyperactivity and stress, and contributed to a more stable mood.


*I see a difference in his anxiety, which has changed with the weighted blanket... It’s gone. It’s almost gone, I can tell you, all day. He is not as worried during the day. He has always been a worrying child and it’s like a rock is somehow lifted... There has been such a difference in his behavior, really in his anxiety. He was worried about or stressed about everything you would do, go to or do. This was concerning for him, but no more. He is basically much calmer. But, as I said, he still needs to know what we are to do and not to do. He is somehow more stable... He is a more secure child, I know. He is calm, as I said, he is not hyperactive, but he used to be much more stressed. So I sense like he is feeling better. No, he is not as worried as he used to be but somewhat more stable.*
(Parent no. 5)

Many of the children had previous experiences of severe anxiety, resulting in anxiety attacks, chest pain, palpitation, cramps, and migraines, for which the families had sought help from both primary and emergency care. The parents noticed that such symptoms decreased or disappeared when their child used the weighted blanket. One explanation for this, provided by the parents, was the improved sleep as the result of the weighted blanket.


*She was kind of getting anxious… She suffered from pain that almost felt like stabbing and you get that in anxiety attacks. We went to the doctor and they pretty much agreed that it was from the anxiety. It’s her head that haunts her, but it’s been a long time since I’ve heard anything about it (since the child started with the weighted blanket). I think there are several different aspects to it. On the one hand, sleep is one aspect, which has made it much better on that front so that she doesn’t have that anxiety in her body. Then the fact that she was diagnosed (with ADHD), and was told by a doctor why she is the way she is. Both things combined have kind of calmed her down to such a degree that she is like a normal person again... it feels like she is much calmer in every way.*
(Parent no. 13)

The parents expressed that their children, through a good sleep with the weighted blanket, got more energy in everyday life, became more positive and happier, and regained their zest for life. Some children had had previous experiences of losing their zest for life and some had suicidal thoughts or had even attempted suicide.


*Lately, I think she has become more alert, I mean happy, she is generally happy but you noticed before when she had slept or been tired that she became more irritable and things like that. But I find it’s positive. She is always happy and less tired.*
(Parent no. 6)

### 3.3. Mastering Everyday Life

Good sleep with a weighted blanket facilitated children’s mastering of everyday life through improved balance in life, family function, and participation in school and leisure activities.

#### 3.3.1. Improved Balance in Life

The parents described an improved balance in life when the children were sleeping with weighted blankets. They could wake up on their own without having to be woken by an alarm or the parents. Being well-rested and alert in the morning allowed the children to have a proper breakfast and to also manage other meals during the day, which had previously been a problem.


*I can tell that she gets up at 6 a.m. every day. And even more rested and that’s where we notice it. She’s nice when she wakes up and has her breakfast.*
(Parent no. 1)

The parents described that the children were more stable in their mood and had greater patience in everyday life because they slept well during the night. The previous mood problems, such as anger, irritability, and rapid mood fluctuations, had a great impact on the parents’ well-being and their relationship with their children. However, when the children had slept well, they were more able to cope in a better way with potential adversities during the day, resulting in fewer and less extensive mood problems. As an example, one mother described that her child was more satisfied with herself and able to deal with unplanned activities, not only those adapted to the child’s interests. The parents also described that better sleep influenced the children by letting them stay focused throughout the day without needing the attention of parents all the time.


*He feels more satisfied with himself. It could be that he sleeps deeper with the blanket, which makes him sleep better. Sleep affects a lot, of course. That he feels more balanced. Before, he could say that he hated himself and he often got angry. I don’t notice that as often anymore. It feels good to have a child who’s feeling better. It’s going in the right direction. It feels good. Just that he enjoys his weighted blanket. It feels good… Now it feels like we are more on track and he feels much more balanced as well and the days are moving on. It is not a battle all the time about what to do or an argument, but we complement each other more at home, even the children. NN can join in on something that I have suggested. It’s not just that you focus and plan around him as it used to be. Everything was prepared so that he would not oppose it or get angry. Now he can take more adversities and when he feels better, it rubs off on everyone.*
(Parent no. 24)

#### 3.3.2. Improved Family Functioning

The parents described that the improvement in their children’s sleep had a positive effect on their family situation. The improvement in sleep resulted in an improved family function for the family, with fewer conflicts, a sense of having more energy among all family members, and that the children were more inclined to do things together with other family members, such as eating together and participating in conversations.


*So much less fuss about his behavior and ADHD symptoms, it’s not at all the same... It’s diminished. It’s a big rock that’s lifted somehow... That’s the magic blanket, yes, it’s magical. I never thought it would affect him so much, I didn’t think it could give him the weight that makes... so it’s really magical. It has made it much easier for us and for him.*
(Parent no. 5)

When the children previously had problems sleeping or slept badly, it resulted in sleep for parents and siblings also being negatively affected. Parents had difficulties sleeping and relaxing in the evenings when they heard their children knocking on the walls, repeatedly getting up and wanting to urinate, have something to eat, or to get into their parents’ bed during the night. This had the effect that the parents were not able to do what they wanted, such as watching movies or just spending time together.


*She’s really sensitive to any noise. I like to sit and watch a movie in the evening, then you have to wear headphones because otherwise she wakes up and gets really irritated. I can watch at any volume now, she lies there completely still, only when she’s wearing the weighted blanket. Otherwise, when she has the normal blanket, she falls back and then it’s like she’s lying around spinning. There’s a huge difference. She sleeps much deeper. Noise and stuff, it doesn’t bother her then.*
(Parent no. 13)

The parents described that a good sleep gave them more energy as parents and an improved ability to cope with problems in everyday life. Previously, the parents had to put a lot of their energy into dealing with the difficulties that came with their children’s anger, or fights between the siblings, which felt manageable when the children slept well. Parents also described that this had previously been so challenging for them that they had been on sick leave to be able to handle everyday life at home.


*I feel like I get more energy when she’s slept well. A lot of my time went into taking on all the fuss with her if she had slept badly and was in a bad mood. Go between her and her siblings, all those squabbles have been a big problem for many years. We’ve been working on it a lot, just that she sleeps well. It makes it incredibly easy, I think... Yes, you start to feel like you can handle a little more yourself. I’ve even been on sick leave. A lot of this bother is one reason I’ve been so tired. It takes a lot of energy and so you have to cope with the rest as well, which takes a lot of energy. When this aspect has disappeared, you get a lot of energy back, so you can deal with other things instead.*
(Parent no. 12)

When the children slept well with weighted blankets, parents described that they could spend time with their children without a lot of friction and manage to collaborate on daily routines. The children got energy to help with everyday things and to learn new things at home, such as cooking and cleaning. The children were more patient, and did not argue as much with their siblings, or get angry when asked to help their parents. Parents also described that siblings played and spent time together in a way that they were not able to before, socializing and having fun together, and talking to each other in a normal way.


*She doesn’t mess as much with her little sister and she can even hang out with her little sister all of a sudden because she’s got better sleep. Otherwise, she is just bothered by everything her little sister does. But when she’s been sleeping well, happy family sort of. I guess it’s us (the family) that it affects the most. You lose energy when the child is affected so much. After all, it sucks all the energy out of you (when the child is not sleeping). Now, you get more help from her. Now, it’s really nice for me at home. I live the super dream life.*
(Parent no. 13)

#### 3.3.3. Improved Participation in School Activities

The parents described that the children’s participation in school had changed for the better. In the past, the children had difficulty coping with a full day at school and staying concentrated and sitting still. This had previously led to difficulties at school and that teachers often called the parents to talk about difficulties and to discuss solutions. When the children started to sleep well, the parents noticed that they received fewer calls from school about these problems. The children started to appreciate school more and even found it fun to go to school and to do homework.


*I don’t think I would have even thought of getting her a blanket like that, because I wouldn’t have believed in it. But just giving her peace and thus can be at school and stuff, that does a lot… it is important for her to sleep her hours at night, in order to manage at school... There aren’t that many days you get a call from school anymore. I think I’ve had one or two calls from school, since she started with this (weighted) blanket. Before, it was pretty often. We’ve even sat down with the school and talked about what we’re going to do to make sure she can cope all day.*
(Parent no. 8)

The parents described that sleeping with weighted blankets increased the children’s participation in both structured and unstructured activities at school, resulting in increased school attendance.


*But she seems more collected and concentrated (when the child has slept well) she doesn’t get these outbursts of anger and leaves all the lessons or leaves school or jumps out of the window or things like that... Yes, given that she has not slept and so she was turning around the clock, she slept in the middle of the day when she should be going to school. We didn’t even get her off to school and now (when she uses the weighted blanket) she’s at school every day... It was terrible before, then we really had to keep going at least for an hour to get her out. Then she could just call from school and say “I’m going home now”. Of course, if you haven’t slept, you have no gusto or energy. You can’t be bothered to eat or to do anything, that’s the way it is. While I can’t say 100 percent that it has to do with the weighted blanket, at least it’s changed drastically. *
(Parent no. 1)

#### 3.3.4. Improved Participation in Leisure Activities

The parents described that the children started spending more time with their friends after school. They had got so much more energy that it allowed them to be active after school. They were out playing, hanging out with friends, and more active in their leisure time. This was a big change, as they previously did not have the energy to spend time with friends after school and mostly lay in bed or played on the computer.


*She sleeps so much better now. That’s why she gets so much energy for the rest of the day. Now you never see her, she is out playing. She is hanging out with friends all the time. A lot more active. It’s awesome, that sleep can affect so much that all of a sudden you have the energy to do more things.*
(Parent no. 13)

The improved participation in leisure activities also included sleepovers with friends. When the children felt comfort in that they could fall asleep on their own, overnight stays with friends also improved.


*From not wanting to sleep over at friends for maybe a year, he slept over for the first time and he brought his blanket.*
(Parent no. 7)

The parents also expressed that a better sleep with a weighted blanket contributed to the children having enough energy left over to do their homework, after having been at school for a full day. This meant that they achieved an inner calm, allowing them to sit still for longer periods and concentrate.


*She can do homework. It barely happened before. That is, just this thing to stay seated. She can sit still. It’s not a problem. She can sit in the same place for half an hour. It’s not a problem.*
(Parent no. 19)

## 4. Discussion

This study explored parents’ experiences of the impact of a non-pharmacological intervention with weighted blankets on sleep among children with ADHD and sleep problems. To the best of our knowledge, this is the first qualitative study investigating the parents’ experiences of the impact of weighted blankets in this context. Overall, the results show that the parents described effects on sleep quality and reduced ADHD symptoms similar to those seen in the few previous intervention studies [32,33] investigating effects of weighted blankets among children with ADHD. However, the parents also expressed that weighted blankets had a positive impact on the children’s overall well-being, as the children were able to relax more easily and their anxiety decreased, while their zest for life returned or increased. Another area where the weighted blankets had an impact was how the children mastered everyday life, resulting in an improved balance in life and family function, as well as in increased participation in school and leisure activities. The experiences expressed by the parents imply that weighted blankets may have positive, indirect effects from a wider perspective, rather than just on sleep problems. The study contributes to an in-depth understanding of the effects of the intervention on their children’s sleep and the consequences this effect has had on their children’s ADHD-related symptoms, their well-being, and their functioning in the family and in everyday life, including in the home environment, leisure time, and school.

### 4.1. The Impact on Sleep Problems

Achieving satisfactory sleep by using weighted blankets was experienced as an essential outcome for children with ADHD and sleep problems. Prolonged sleep onset latency was expressed by most parents as a significant reason why the children had sleep problems, and this was reduced when using the weighted blanket. There have been a few quantitative studies, which indicate that the use of weighted blankets significantly reduces sleep onset latency in a range from fewer than 20 min (measured by actigraph) [33], to 30 min (parent-reported), among children with ADHD [32,33]. The effect on sleep onset latency is of significant importance, given that problems with prolonged sleep onset latency are more common in children with ADHD than among healthy peers [32,33]. About one-third of children with ADHD have difficulties falling asleep [3,44], with up to one hour of sleep onset latency between children with ADHD and healthy peers [7].

The parents in our study also experienced an improvement in sleep continuity and depth, with fewer awakenings during the night. When the children slept more calmly, deeply, and continuously, they received more hours of night sleep than before. These improvements in sleep duration, sleep quality, and nightly awakenings described by the parents are, however, not confirmed by previous studies measuring the effect of weighted blankets among children with ADHD [32] and autism [32,34]. The discrepancy between what the parents in our current study experienced and what has been seen in those small, controlled studies investigating measurable effects may be due to aspects related to, e.g., selection of study participants, or the size of the intervention and control groups. It could also be that the parents experienced improvements in sleep that were not captured by the instruments and objective measurements used in the previous quantitative studies. Further research is needed to better understand if and how weighted blankets could increase sleep continuity and sleep depth and how this contributes to health-related outcomes.

The parents in our study expressed how sleep routines are of importance for the children, and how the weighted blankets could serve as an aid in creating these routines around bedtime. Sleep routines are described as a set of behavioral, environmental, and cognitive modifications that need to be made to improve sleep, and it seems as if the use of the weighted blanket helped the children to make some of these modifications. One example of this was how placing the weighted blanket on the child served as a sign for bedtime. For the children, the weighted blanket contributed to a feeling of comfort and a desire to stay in bed or even acted as an obstacle to getting out of bed, thereby facilitating sleep. Another example was that the children fell asleep in their own bed, at a distance from the parents’ bed. Thus, weighted blankets may serve as a part of a sleep intervention to improve sleep routines for these children. The benefits of improved sleep routines for this target group were confirmed in a systematic review [45] and are important because children with ADHD more often show bedtime resistance, compared to healthy peers [7,46]. Since children are closely tied to their parents in relation to sleep routines, modifications (behavioral, environmental, and cognitive) have to be designed together with both parents and children to meet their needs and preferences [21,22].

### 4.2. The Impact on Overall Well-Being

Experiences of effects related to overall well-being were another central finding in our study. The parents described that the children’s well-being increased, both when they were going to sleep and lying in bed, but also during the day, in the form of decreased anxiety and increased feelings of happiness. The children had expressed that the weighted blankets gave them a feeling of calmness and that they felt embraced by the blanket. They expressed it as if they felt hugged and calm when they slept with it. Similar positive experiences have previously been presented, where children with autism “really liked” the weighted blanket and their parents felt that sleep was improved and that their child was calmer with it [34]. Overall well-being was also influenced when anxiety improved with the weighted blankets. Reduced anxiety is an important finding in our study, as anxiety is a significant and common problem for about half of all children with ADHD [47] and an important factor associated with sleep problems [9]. The consequence of reducing anxiety with weighted blankets has therefore dual advantages—both symptom relief and improved sleep. The parents in our study described how, after a good night’s sleep with the weighted blanket, the children were more positive and happy, showing regained joy of life.

### 4.3. The Impact on Everyday Life

The findings from our study show that, when children woke up well-rested in the morning, they had an improved ability to cope with everyday life and the routines and challenges involved. This adds value to previous research that showed an improved level of functioning among children with ADHD sleeping with weighted blankets [33]. The parents in our study expressed that, when children’s sleep was improved, it infused the family life with harmony; there were fewer conflicts and behavioral problems and improved family function. Results from an intervention study with weighted blankets among children with ADHD support this finding [33]. There is a great need to offer children sleep interventions, such as with a weighted blanket, since earlier research has shown that sleep problems are associated with a significant negative impact on children’s functioning and quality of life, after controlling for ADHD symptoms [48]. The reduction of behavioral problems is, in itself, a positive outcome of our intervention, but has an additional effect on improving sleep through diminishing the negative relationship between behavioral problems and sleep problems and the potential exacerbation of sleep problems in children with ADHD, as described in a systematic review by Bondopadhyay et al. (2021). This suggests that the reduction of behavioral problems is of great importance, both as an outcome to evaluate in clinical settings as well as in research studies, in conjunction with sleep interventions for this target group.

A further finding in our study was that the parents described how family functioning improved; children’s improved sleep contributed to better sleep for the parents as well, which made them better equipped for the challenges of everyday life. Previous research demonstrated that families of children with autism spectrum disorder that have good sleep were also more resilient [49].

Another benefit of improved sleep was increased participation in both structured and unstructured school activities and improved coping with school work. Hvolby (2020) has previously reported fewer behavior problems, especially in relation to school, when children with ADHD use weighted blankets [33], which may be one reason for the increased participation at school seen in our study. Persistent sleep problems among children with ADHD are known to be associated with impaired academic performance [50,51,52], and impaired child–teacher relationships and greater conflicts between the child and the teacher [16]. Altogether, this highlights the importance of improving sleep among children with ADHD from the perspective of school participation and achievement and emphasizes the potential of providing interventions with weighted blankets for sleep problems at an early stage.

### 4.4. Methodological Considerations

In qualitative research, trustworthiness is defined according to the four criteria of credibility, dependability, confirmability, and transferability [37,53,54]. *Credibility* refers to confidence in the truth of the data and the analysis [37,54]. In this study, the credibility was strengthened by a purposeful sample of 24 parents of 12 girls and 12 boys with ADHD and sleep problems. There was variation in terms of sex, age, civil status, education, employment, and place of residence. However, the included parents represent to a great extent parents who have expressed positive effects of weighted blankets for their child’s sleep at 8-week follow-up. Nevertheless, some of the parents interviewed had not seen a great effect from weighted blankets. A limitation is that parents of children who chose the control blanket for the 8-week follow-up had been excluded from this study. However, the selection of participants is considered appropriate since the purpose was to understand how parents experienced the long-term effects of weighted blankets and why they had chosen the weighted blanket for the 8-week-follow-up. To avoid memory decay over time, all interviews were conducted immediately when the intervention ended; despite this, some parents stated that they had forgotten the initial effect of the weighted blanket. Credibility was strengthened by the researchers’ familiarity with the methodology, the careful descriptions of the data collection and analysis, and the continuous discussion between researchers during the analysis. *Dependability* refers to the stability of data over time and conditions [37,54]. Dependability was strengthened by the fact that the same researcher conducted all the interviews, and that all interviews began with the same opening question, follow-up questions were posed to avoid misunderstanding, and the participants were encouraged to talk openly. *Confirmability* refers to the neutrality of the data, which ensures that the data represent the information provided by the participants and accurately reflect their voices [54]. Confirmability was demonstrated through all steps of the analysis, which have been carefully reported and that the participants’ experiences are described in as much detail as possible, with quotations that enhance and illuminate the content. The interviews were rich in content and contained a great variety of experiences. *Transferability* refers to the applicability of the results to other contexts [37,54]. A strength of this study is that it included parents of children who were recruited from a clinic in conjunction with recently diagnosed ADHD and sleep problems. The study was part of a larger RCT study to recruit children requiring a sleep intervention in a clinical setting. The experiences expressed in this study are thus likely to be similar to those of parents in a clinical setting and the results are thus possibly transferable to children with other neurodevelopmental syndromes.

## 5. Conclusions

This study provides insights from the experiences of parents of children with ADHD and sleep problems who participated in an intervention with weighted blankets. When using weighted blankets, the children achieved satisfactory sleep, including improved sleep onset latency, sleep continuity, and sleep routines. The overall well-being among the children was also improved, with increased relaxation, reduced anxiety, and increased joy of life. The improved sleep improved their mastering of daily life, through better balance in life, family function, and participation in school and leisure activities, which is partially in line with previous intervention studies. However, this study brings forward new aspects of the effects of improved sleep among children with ADHD. These aspects should be included in the evaluation of weighted blankets, but also contribute to the overall understanding of the potential positive effects. Our findings reveal that using weighted blankets can improve the well-being and life of children with ADHD and their families.

## Figures and Tables

**Table 1 ijerph-18-12959-t001:** Sociodemographic data of parents to children preferring weighted blankets (*n* = 24).

Variable	Parents (*n* = 24)
Sex, female/male (*n*)	18/6
Age in years, median (range)	39 (32–55)
Civil status, co-habiting/living alone (*n*)	20/4
Educational level, primary school/secondary/university (*n*)	2/10/12
Employment, full-time/part-time/unemployed/sick leave (*n*)	15/7/1/1
Native-born/Foreign-born (*n*)	22/2
Place of residence, city/countryside (*n*)	8/16
Age of the child in years, median (range)	9 (6–15)
Sex of the child, female/male (*n*)	12/12
Perceived effect of the weighted blanket, fully/partially/no (*n*)	19/3/2

**Table 2 ijerph-18-12959-t002:** Overview of the categories and subcategories showing parents’ experiences of the weighted blankets’ impact on children with ADHD.

Categories	Subcategories
Achieving satisfactory sleep	Improved sleep onset latency
	Improved sleep continuity
	Improved sleep routines
Experiencing overall well-being	Improved relaxation
	Reduced anxiety
Mastering everyday life	Improved balance in life
	Improved family functioning
	Improved participation in school activities
	Improved participation in leisure activities

## Data Availability

Not applicable. The data will not be shared, because the ethics approval for the study requires that the transcribed interviews be kept in locked files, accessible only to the researchers.

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
