# Peer review of "Parents’ Experiences of Weighted Blankets’ Impact on Children with Attention-Deficit/Hyperactivity Disorder (ADHD) and Sleep Problems—A Qualitative Study"

_ijerph, 2021, doi:10.3390/ijerph182412959_

Round 1

Reviewer 1 Report

I am glad about this opportunity to review this paper. Thanks for submitting this paper “Parents’ experiences of the weighted blankets’ impact on children with attention-deficit/hyperactivity disorder (ADHD) and sleep problems – an interview study in the SLEEP intervention”. This qualitative study examined the effectiveness of weighted blankets in addressing sleep disturbances among children with attention deficit and hyperactivity disorder.

The authors must be commended for the following aspects of their manuscript:

  • The topic is very interesting and addresses a major problem for students with ADHD.
  • Excellent literature review, the gaps in the previous study well explored and discussed with relevant citations and references.
  • The intervention was clearly described.
  • The recruitment of the participants was clearly explained.
  • Data collection procedure, data analysis, ethical consideration, result, and discussion sections were all well-written and adequate.

I think that the article is well written and very relevant and must be accepted for publication. I only have these minor comments. These are just for you to consider. 

Topic

I think that the topic is too long and I recommend that the sub-title should rather describe the research design adopted for the study - Parents’ experiences of the weighted blankets’ impact on children with attention-deficit/hyperactivity disorder (ADHD) and sleep problems – a randomized controlled study.

Abstract

Line 16 mentioned, “Controlled studies”, can you please clarify if this current study is a controlled study? If so, then I think the purpose of your intervention should have compared two groups – one using a weighted blanket and another not using a weighted blanket. In addition, you need to make this clear in the abstract by making a comparison with the controlled group.

In addition, your abstract is silent on how the 24 parents were recruited for the study. Can you please briefly say something about how the participants were recruited for the study?

Introduction

2.3. Participants

Although, you have explained that you used a purposive sampling approach it will be helpful to your readers to also provide some clear criteria of inclusion of the participants you interviewed. What made them more qualified for the interview?

Author Response

Review 1:

I am glad about this opportunity to review this paper. Thanks for submitting this paper “Parents’ experiences of the weighted blankets’ impact on children with attention-deficit/hyperactivity disorder (ADHD) and sleep problems – an interview study in the SLEEP intervention”. This qualitative study examined the effectiveness of weighted blankets in addressing sleep disturbances among children with attention deficit and hyperactivity disorder.

The authors must be commended for the following aspects of their manuscript:

  • The topic is very interesting and addresses a major problem for students with ADHD.
  • Excellent literature review, the gaps in the previous study well explored and discussed with relevant citations and references.
  • The intervention was clearly described.
  • The recruitment of the participants was clearly explained.
  • Data collection procedure, data analysis, ethical consideration, result, and discussion sections were all well-written and adequate.

I think that the article is well written and very relevant and must be accepted for publication.

Answer: Thank you very much!

I only have these minor comments. These are just for you to consider. 

Topic

I think that the topic is too long and I recommend that the sub-title should rather describe the research design adopted for the study - Parents’ experiences of the weighted blankets’ impact on children with attention-deficit/hyperactivity disorder (ADHD) and sleep problems – a randomized controlled study.

Answer: Thank you for the suggestion. We have changed the title: “Parents’ experiences of the weighted blankets’ impact on children with attention-deficit/hyperactivity disorder (ADHD) and sleep problems– a qualitative study”

Abstract

Line 16 mentioned, “Controlled studies”, can you please clarify if this current study is a controlled study? If so, then I think the purpose of your intervention should have compared two groups – one using a weighted blanket and another not using a weighted blanket. In addition, you need to make this clear in the abstract by making a comparison with the controlled group.

Answer: Thank you for noticing this, of course this is not a controlled study. We have changed the sentence to: “Research on experiences of weighted blankets and their effect in sleep improvement is scarce.”

In addition, your abstract is silent on how the 24 parents were recruited for the study. Can you please briefly say something about how the participants were recruited for the study?

Answer: Thank you for noticing this. We have changed the sentence to: “Interviews were conducted with a purposeful sample of 24 parents of children with ADHD and sleep problems, after completing a sleep intervention with weighted blankets”

Introduction

2.3. Participants

Although, you have explained that you used a purposive sampling approach it will be helpful to your readers to also provide some clear criteria of inclusion of the participants you interviewed. What made them more qualified for the interview?

Answer: Thank you for the suggestion. We have changed the sentence to: “The participants were strategical selected to achieve variation in sex, age, civil status, educational level, employment, place of residence.”

Reviewer 2 Report

This study is well described and written with some information requiring clarification and minor re-writing.

Although it’s a qualitative study, it employed an intervention of a weighted blanket and a control blanket. The robustness of the study can be significantly improved by randomisation of the weighted and control blankets; and by interviewing the parents twice, once after 4 weeks of weighted blanket and once after 4 weeks of control blanket to allow direct contrast in experiences and avoiding memory decay over time. The authors may consider these as study limitations.

The application of the phrase "evidence-based practice" in this paper needs to be avoided. Qualitative research has not been considered sound evidence for practice. In fact, the level of evidence for a single qualitative study is assigned Level VI, with evidence from a systematic review or meta-analysis of all relevant RCTs assigned Level I. However, this qualitative study provides important insight into understanding the positive effects of weighted blankets on the sleep and wellbeing of children with ADHD.

Specific comments:

ABSTRACT

Line 21: Please change “experienced” to “reported”

Line 21: Please provide a brief, concise description of Methods by clarify how long for which the weighted blanket and control blanket was applied and indicate a set of standard questions used.

Line 27: Please delete "and support their use in" and replace with "critical for"

INTRODUCTION

Line 59: Please indicate what type of side effects briefly.

Line 62: Please provide a brief account of the rationale for using weighted blankets. Why chose this over other blanket types?

Line 73-76: Suggest a re-write of this sentence, for example, "In filling this gap, an RCT by our research group is underway...."

This new sentence will give the statement currency...

Line 79-81: Please either delete this statement or re-write it to emphasise the qualitative aspects (e.g., experiences, thoughts, etc) of study on this topic.

Line 85: spelling error “stydy”

MATERIALS AND METHODS

Line 98: Who made the diagnosis?

Line 101: What type of fibres was used to increase the weight of the weighted blanket? How was the weight of the blanket determined? Was there a formula of some kind?

Lines 105-106: Was the order of the weighted and control blanket randomised?

Line 106: please delete “these” eight…

Line 117: Data collection

- Were the interviews done at the end of the 8 weeks of study intervention?

- The comments by some parents suggested that they did not quite remember the experiences voiced by the children.

- The parents’ comments also suggested that the weighted blankets were used in the first 4 weeks followed by the lighter blanket.

If so, please discuss these as study limitations.

Line 138: Data analysis

As part of COREQ:

Please clarify what software, if applicable, was used to manage the data.

Please indicate if participants provided feedback on the findings.

RESULTS

Line 263: Please indicate the proportion of parents interviewed who had not seen a great effect from weighted blankets.

DISCUSSION

Line 542: Please replace "evidence-based practice" with another term/ phrase OR delete this sentence. The sentence that follows suffices.

Line 649: Need mention here that interviews should have been held for ‘after the weighted blanket’ and ‘after control blanket’ to capture the differences in effects.

Lines 652-654: This statement should be removed.

CONCLUSIONS

Line 684: Please re-write this phrase “Our findings support the use of weighted blankets for ADHD” because this qualitative study (at level VI evidence) does not support evidence-based practice.

Author Response

Review 2:

This study is well described and written with some information requiring clarification and minor re-writing.

Although it’s a qualitative study, it employed an intervention of a weighted blanket and a control blanket. The robustness of the study can be significantly improved by randomisation of the weighted and control blankets; and by interviewing the parents twice, once after 4 weeks of weighted blanket and once after 4 weeks of control blanket to allow direct contrast in experiences and avoiding memory decay over time. The authors may consider these as study limitations.

Answer: The study is robust with randomisation of the weighted and control blankets. The study also has a cross-over design. We have added a sentence to clarify that the interviews were conducted immediately when the intervention ended and the children had used the weighted blanket for the past 8 weeks: “However, the selection of participants is considered appropriate since the purpose was to understand how parents experienced the long-term effects of weighted blankets and why they had chosen the weighted blanket for the 8 week-follow-up. To avoid memory decay over time, all interviews were conducted immediately when the intervention ended, despite this, some parents stated that they had forgotten the initial effect of the weighted blanket.” Page 14

The application of the phrase "evidence-based practice" in this paper needs to be avoided. Qualitative research has not been considered sound evidence for practice. In fact, the level of evidence for a single qualitative study is assigned Level VI, with evidence from a systematic review or meta-analysis of all relevant RCTs assigned Level I. However, this qualitative study provides important insight into understanding the positive effects of weighted blankets on the sleep and wellbeing of children with ADHD.

Answer: Thank you for noticing this. In the introduction we have deleted one sentence about evidence-base practice Page 2. In the discussion we have deleted one sentence about evidence-base practice Page 12. We have changed the sentence: “This type of knowledge is important, both to provide understanding of the intervention as such, and in the development of sustainable interventions using weighted blankets for sleep improvement in clinical practice” Page 2

Specific comments:

ABSTRACT

Line 21: Please change “experienced” to “reported”

Answer: Thank you for the suggestion. We have changed the sentence to: “Parents reported that children sleeping with weighted blankets”

Line 21: Please provide a brief, concise description of Methods by clarify how long for which the weighted blanket and control blanket was applied and indicate a set of standard questions used.

Answer: Thank you for the suggestion. We would like to add this to the abstract but due to the limited number of words (n = 200) we can only mention that the intervention is 16 weeks. This is added to the abstract:” Interviews were conducted with a purposeful sample of 24 parents of children with ADHD and sleep problems, after completing a sleep intervention with weighted blankets for 16 weeks.”

Line 27: Please delete "and support their use in" and replace with "critical for"

Answer: Thank you for the suggestion. We have changed the sentence to: “The findings contribute to the understanding of potential positive effects of an intervention with weighted blankets critical for clinical practice to improve sleep, wellbeing, and everyday life of children with ADHD and their families.”

INTRODUCTION

Line 59: Please indicate what type of side effects briefly.

Answer:  Thank you for the suggestion. We have added a sentence about this:” Melatonin is the most common medication for sleep in various neurological conditions and is regarded as safe and effective [26] but might cause morning drowsiness, increased enuresis, headache, dizziness, diarrhea, rash, and hypothermia [27].” Page 2

Line 62: Please provide a brief account of the rationale for using weighted blankets. Why chose this over other blanket types?

Answer: Thank you for the suggestion. We have added two sentences about this: “The use of weighted blankets is based on the theory of sensory integration with assumptions of how the body reacts to different stimuli depending on the sensitivity to stimuli [30,31]. A weighted blanket is considered a sensory-stimulating, cognitive aid that provides a deep pressure on the body (Deep Pressure Touch Stimulation). The pressure created by the weight from the weighted blanket leads to a feeling of being enveloped and increasing body awareness [32].”

Line 73-76: Suggest a re-write of this sentence, for example, "In filling this gap, an RCT by our research group is underway...."

This new sentence will give the statement currency...

Answer: Thank you for the suggestion. We have changed the sentences: “In filling this gap, an RCT by our research group is underway to explore the effects of an intervention with weighted blankets on both sleep and everyday life for children with ADHD and sleep problems.”

Line 79-81: Please either delete this statement or re-write it to emphasise the qualitative aspects (e.g., experiences, thoughts, etc) of study on this topic.

Answer: we have deleted the statement

Line 85: spelling error “stydy”

Answer: Thank you for noticing this.

MATERIALS AND METHODS

Line 98: Who made the diagnosis?

Answer: The child attended a special ADHD team. We have added some sentences about this: “Diagnoses were based on a detailed written report from school including a SNAP-IV rating scale and an interview with parents and the child performed by a registrar or locum junior doctor utilizing a structured protocol. The decision was always discussed and confirmed by a consultant child psychiatrist during the diagnostic session. The consultant could choose to interview the child and parents as needed but an oral report was mandatory and mostly sufficient.. Page 3

Line 101: What type of fibres was used to increase the weight of the weighted blanket? How was the weight of the blanket determined? Was there a formula of some kind?

Answer: We have added a sentence about the type of fibres: “The fibers used in the weighted blanket consist of polyester filaments.” Page 3
The weight of the blanket was determined by the experienced occupational therapists based on age, sex, height, weight, degree of sleep problems, and subtypes of ADHD, in accordance with the use in the clinical practice of two We have stated a sentence about this: “The weight of each blanket was individually tailored (weight between 6 to 10 kg) for the children based on age, sex, height, weight, degree of sleep problems, and subtypes of ADHD, in accordance with the use in the clinical practice of two, independent, experienced occupational therapists” Page 3

Lines 105-106: Was the order of the weighted and control blanket randomised?

Answer: Yes, the order of the weighted and control blanket randomized. We have clarified this “The children were randomized to start either with a weighted blanket or with a control blanket and used each blanket for four weeks and then changed to the other blanket (a cross-over design).”Page 3

Line 106: please delete “these” eight…

Answer: Thank you for the suggestion. We have deleted “these”

Line 117: Data collection

- Were the interviews done at the end of the 8 weeks of study intervention?

Answer: The interviews were conducted after 16 weeks. First the randomized part, 4 weeks with a weighted blanket + 4 weeks with a control blanket + 8 weeks with a preferred blanket (weighted or control blanket). Under the section participants, we have stated: “Twenty-six parents of the children in the intervention study were approached at the end of the 16-week intervention and asked to participate in an interview about their experiences of using weighted blankets. A total of 24 parents accepted the invitation and participated.” Do you think we need to clarify this further?

- The comments by some parents suggested that they did not quite remember the experiences voiced by the children.

- The parents’ comments also suggested that the weighted blankets were used in the first 4 weeks followed by the lighter blanket.

If so, please discuss these as study limitations.

Answer: The interviews were conducted after 16 weeks. First the randomized part, 4 weeks with a weighted blanket + 4 weeks with a control blanket + 8 weeks with a preferred blanket (weighted or control blanket). We have tried to clarify the study limitation regarding this: “ However, the selection of participants is considered appropriate since the purpose was to understand how parents experienced the long-term effects of weighted blankets and why they had chosen the weighted blanket for the 8 week-follow-up. To avoid memory decay over time, all interviews were conducted immediately when the intervention ended, despite this, some parents stated that they had forgotten the initial effect of the weighted blanket.” Page 14

Line 138: Data analysis

As part of COREQ:

Please clarify what software, if applicable, was used to manage the data.

Answer: Thank you for noticing this. No software was used to manage data.

Please indicate if participants provided feedback on the findings.

Answer: Thank you for noticing this. The participants had not been provided feedback on the findings

RESULTS

Line 263: Please indicate the proportion of parents interviewed who had not seen a great effect from weighted blankets.

Answer: In qualitative research, we do not usually mention the proportions of participants. But in table 1 you can see the number of parents who perceived fully, partially or no effect of the weighted blanket

DISCUSSION

Line 542: Please replace "evidence-based practice" with another term/ phrase OR delete this sentence. The sentence that follows suffices.

Answer: Thank you for noticing this. We have deleted the sentence about evidence-base practice Page 12.

Line 649: Need mention here that interviews should have been held for ‘after the weighted blanket’ and ‘after control blanket’ to capture the differences in effects.

Answer: In our study, we focus on the experiences of the weighted blanket but it is an interesting thought to capture the differences in effects between the blankets.

Lines 652-654: This statement should be removed.

Answer: Thank you for noticing this. We have deleted the sentence  

CONCLUSIONS

Line 684: Please re-write this phrase “Our findings support the use of weighted blankets for ADHD” because this qualitative study (at level VI evidence) does not support evidence-based practice.

Answer: Thank you for noticing this. We have re-written the sentence: “Our findings reveal that using weighted blankets can improve the wellbeing and life of children with ADHD and their families”.

Reviewer 3 Report

This is very interesting study. However it has a few flaws which should be corrected:

  1. Data collection; page 3; Authors have to provide the full name of the University where have been interviews performed.
  2. Authors have to provide detailed inclusion and exclusion criteria for participants selection in Materials and Methods section.
  3. Authors have to move sociodemographic data (Table 1) to the Results section.
  4. Authors have to extend a description of interview questions preparation and justify why such questions were applied. Readers should know the reliability of applied questions.

Author Response

Reviewer 3:

This is very interesting study. However it has a few flaws which should be corrected:

  1. Data collection; page 3; Authors have to provide the full name of the University where have been interviews performed.

Answer: Thank you for noticing this. We add Halmstad University

  1. Authors have to provide detailed inclusion and exclusion criteria for participants selection in Materials and Methods section.

Answer: The inclusion criteria were that the participant would be a parent of a child with ADHD and sleep disorders who completed the RCT study and that the child had used the weight blanket at the 8-week follow-up. “A purposeful sample was carried out among the parents in the SLEEP intervention. The inclusion criteria were: a parent of a child with ADHD and sleep problems, the child had completed the RCT, the child chose to continue with the weighted blanket for the 8 week-follow-up. The participants were strategically selected to achieve variation in sex, age, civil status, educational level, employment, place of residence”.Page 3

  1. Authors have to move sociodemographic data (Table 1) to the Results section.

Answer: We don’t agree. In qualitative studies is the sociodemographic data not a result of the study but a description of the included participants. In qualitative studies, this is usually described in the method section and not in the result.

  1. Authors have to extend a description of interview questions preparation and justify why such questions were applied. Readers should know the reliability of applied questions.

Answer: In qualitative studies, we do not use a validated instrument with questions. The questions are open and focus on the phenomenon under study. The participants are encouraged to describe their experiences in this case: “experiences of weighted blankets for children with ADHD and sleep problems, and its impact on their children’s sleep” We have added a sentence to clarify this: “Qualitative content analysis is useful for analyzing an individual's experiences, reflections or attitudes” Page 3

Round 2

Reviewer 3 Report

I don't have further comments. Good job!